# Evaluating Protein Transfer Learning with TAPE

**Roshan Rao***
UC Berkeley
roshan_rao@berkeley.edu

**Nicholas Bhattacharya***
UC Berkeley
nick_bhat@berkeley.edu

**Neil Thomas***
UC Berkeley
nthomas@berkeley.edu

**Yan Duan**
covariant.ai
rocky@covariant.ai

**Xi Chen**
covariant.ai
peter@covariant.ai

**John Canny**
UC Berkeley
canny@berkeley.edu

**Pieter Abbeel**
UC Berkeley
pabbeel@berkeley.edu

**Yun S. Song**
UC Berkeley
yss@berkeley.edu

## Abstract

Machine learning applied to protein sequences is an increasingly popular area of research. Semi-supervised learning for proteins has emerged as an important paradigm due to the high cost of acquiring supervised protein labels, but the current literature is fragmented when it comes to datasets and standardized evaluation techniques. To facilitate progress in this field, we introduce the Tasks Assessing Protein Embeddings (TAPE), a set of five biologically relevant semi-supervised learning tasks spread across different domains of protein biology. We curate tasks into specific training, validation, and test splits to ensure that each task tests biologically relevant generalization that transfers to real-life scenarios. We benchmark a range of approaches to semi-supervised protein representation learning, which span recent work as well as canonical sequence learning techniques. We find that self-supervised pretraining is helpful for almost all models on all tasks, more than doubling performance in some cases. Despite this increase, in several cases features learned by self-supervised pretraining still lag behind features extracted by state-of-the-art non-neural techniques. This gap in performance suggests a huge opportunity for innovative architecture design and improved modeling paradigms that better capture the signal in biological sequences. TAPE will help the machine learning community focus effort on scientifically relevant problems. Toward this end, all data and code used to run these experiments are available at https://github.com/songlab-cal/tape.

## 1  Introduction

New sequencing technologies have led to an explosion in the size of protein databases over the past decades. These databases have seen exponential growth, with the total number of sequences doubling every two years [1]. Obtaining meaningful labels and annotations for these sequences requires significant investment of experimental resources, as well as scientific expertise, resulting in an exponentially growing gap between the size of protein sequence datasets and the size of annotated subsets. Billions of years of evolution have sampled the portions of protein sequence space that are relevant to life, so large unlabeled datasets of protein sequences are expected to contain significant biological information [2–4]. Advances in natural language processing (NLP) have shown that self-supervised learning is a powerful tool for extracting information from unlabeled sequences [5–7],

which raises a tantalizing question: can we adapt NLP-based techniques to extract useful biological information from massive sequence datasets?

To help answer this question, we introduce the Tasks Assessing Protein Embeddings (TAPE), which to our knowledge is the first attempt at systematically evaluating semi-supervised learning on protein sequences. TAPE includes a set of five biologically relevant supervised tasks that evaluate the performance of learned protein embeddings across diverse aspects of protein understanding.

We choose our tasks to highlight three major areas of protein biology where self-supervision can facilitate scientific advances: structure prediction, detection of remote homologs, and protein engineering. We constructed data splits to simulate biologically relevant generalization, such as a model's ability to generalize to entirely unseen portions of sequence space, or to finely resolve small portions of sequence space. Improvement on these tasks range in application, including designing new antibodies [8], improving cancer diagnosis [9], and finding new antimicrobial genes hiding in the so-called "Dark Proteome": tens of millions of sequences with no labels where existing techniques for determining protein similarity fail [10].

We assess the performance of three representative models (recurrent, convolutional, and attention-based) that have performed well for sequence modeling in other fields to determine their potential for protein learning. We also compare two recently proposed semi-supervised models (Bepler et al. [11], Alley et al. [12]). With our benchmarking framework, these models can be compared directly to one another for the first time.

We show that self-supervised pretraining improves performance for almost all models on all down-stream tasks. Interestingly, performance for each architecture varies significantly across tasks, highlighting the need for a multi-task benchmark such as ours. We also show that non-deep alignment-based features [13–16] outperform features learned via self-supervision on secondary structure and contact prediction, while learned features perform significantly better on remote homology detection.

Our results demonstrate that self-supervision for proteins is promising but considerable improvements need to be made before self-supervised models can achieve breakthrough performance. All code and data for TAPE are publically available[1], and we encourage members of the machine learning community to participate in these exciting problems.

## 2 Background

### 2.1 Protein Terminology

Proteins are linear chains of amino acids connected by covalent bonds. We encode amino acids in the standard 25-character alphabet, with 20 characters for the standard amino acids, 2 for the non-standard amino acids selenocysteine and pyrrolysine, 2 for ambiguous amino acids, and 1 for when the amino acid is unknown [1, 17]. Throughout this paper, we represent a protein $x$ of length $L$ as a sequence of discrete amino acid characters $(x_1, x_2, \ldots, x_L)$ in this fixed alphabet.

Beyond its encoding as a sequence $(x_1, \ldots, x_L)$, a protein has a 3D molecular structure. The different levels of protein structure include *primary* (amino acid sequence), *secondary* (local features), and *tertiary* (global features). Understanding how primary sequence folds into tertiary structure is a fundamental goal of biochemistry [2]. Proteins are often made up of a few large *protein domains*, sequences that are evolutionarily conserved, and as such have a well-defined fold and function.

Evolutionary relationships between proteins arise because organisms must maintain certain functions, such as replicating DNA, as they evolve. Evolution has selected for proteins that are well-suited to these functions. Though structure is constrained by evolutionary pressures, sequence-level variation can be high, with very different sequences having similar structure [18]. Two proteins that share a common evolutionary ancestor are called *homologs*. Homologous proteins may have very different sequences if they diverged in the distant past.

Quantifying these evolutionary relationships is very important for preventing undesired information leakage between data splits. We mainly rely on *sequence identity*, which measures the percentage of exact amino acid matches between aligned subsequences of proteins [19]. For example, filtering at a 25% sequence identity threshold means that no two proteins in the training and test set have greater

than 25% exact amino acid matches. Other approaches besides sequence identity filtering also exist, depending on the generalization the task attempts to test [20].

## 2.2 Modeling Evolutionary Relationships with Sequence Alignments

The key technique for modeling sequence relationships in computational biology is alignment [13, 16, 21, 22]. Given a database of proteins and a query protein at test-time, an alignment-based method uses either carefully designed scoring systems [21] or Hidden Markov Models (HMMs) [16] to align the query protein against all proteins in the database. Good alignments give information about local perturbations to the protein sequence that may preserve, for example, function or structure. The distribution of aligned residues at each position is also an informative representation of each residue that can be fed into downstream models.

## 2.3 Semi-supervised Learning

The fields of computer vision and natural language processing have been dealing with the question of how to learn from unlabeled data for years [23]. Images and text found on the internet generally lack accompanying annotations, yet still contain significant structure. Semi-supervised learning tries to jointly leverage information in the unlabeled and labeled data, with the goal of maximizing performance on the supervised task. One successful approach to learning from unlabeled examples is *self-supervised learning*, which in NLP has taken the form of next token prediction [5], masked token prediction [6], and next sentence classification [6]. Analogously, there is good reason to believe that unlabelled protein sequences contain significant information about their structure and function [2, 4]. Since proteins can be modeled as sequences of discrete tokens, we test both next token and masked token prediction for self-supervised learning.

# 3 Related Work

The most well-known protein modeling benchmark is the Critical Assessment of Structure Prediction (CASP) [24], which focuses on structure modeling. Each time CASP is held, the test set consists of new experimentally validated structures which are held under embargo until the competition ends. This prevents information leakage and overfitting to the test set. The recently released ProteinNet [25] provides easy to use, curated train/validation/test splits for machine learning researchers where test sets are taken from the CASP competition and sequence identity filtering is already performed. We take the contact prediction task from ProteinNet. However, we believe that structure prediction alone is not a sufficient benchmark for protein models, so we also use tasks not included in the CASP competition to give our benchmark a broader focus.

Semi-supervised learning for protein problems has been explored for decades, with lots of work on kernel-based pretraining [26, 27]. These methods demonstrated that semi-supervised learning improved performance on protein network prediction and homolog detection, but couldn't scale beyond hundreds of thousands of unlabeled examples. Recent work in protein representation learning has proposed a variety of methods that apply NLP-based techniques for transfer learning to biological sequences [11, 12, 28, 29]. In a related line of work, Riesselman et al. [30] trained Variational Auto Encoders on aligned families of proteins to predict the functional impact of mutations. Alley et al. [12] also try to combine self-supervision with alignment in their work by using alignment-based querying to build task-specific pretraining sets.

Due to the relative infancy of protein representation learning as a field, the methods described above share few, if any, benchmarks. For example, both Rives et al. [29] and Bepler et al. [11] report transfer learning results on secondary structure prediction and contact prediction, but they differ significantly in test set creation and data-splitting strategies. Other self-supervised work such as Alley et al. [12] and Yang et al. [31] report protein engineering results, but on different tasks and datasets. With such varied task evaluation, it is challenging to assess the relative merits of different self-supervised modeling approaches, hindering efficient progress.

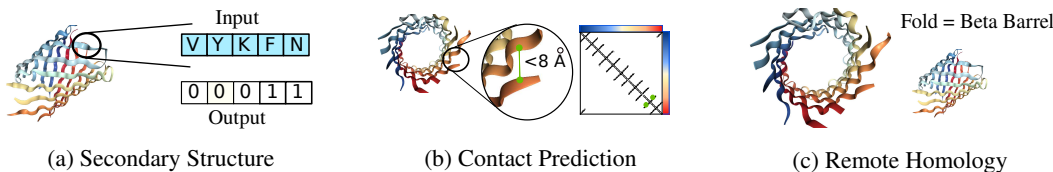

|  | Input |  | Fold = Beta Barrel |
|---|---|---|---|
| (a) Secondary Structure | (b) Contact Prediction | (c) Remote Homology | |

Figure 1: Structure and Annotation Tasks on protein KgdM Porin (pdbid: 4FQE). (a) Viewing this Porin from the side, we show secondary structure, with the input amino acids for a segment (blue) and corresponding secondary structure labels (yellow and white). (b) Viewing this Porin from the front, we show a contact map, where entry $i, j$ in the matrix indicates whether amino acids at positions $i, j$ in the sequence are within 8 angstroms of each other. In green is a contact between two non-consecutive amino acids. (c) The fold-level remote homology class for this protein.

## 4 Datasets

Here we describe our unsupervised pretraining and supervised benchmark datasets. To create benchmarks that test generalization across large evolutionary distances and are useful in real-life scenarios, we curate specific training, validation, and test splits for each dataset. Producing the data for these tasks requires significant effort by experimentalists, database managers, and others. Following similar benchmarking efforts in NLP [32], we describe a set of citation guidelines in our repository[2] to ensure these efforts are properly acknowledged.

### 4.1 Unlabeled Sequence Dataset

We use Pfam [33], a database of thirty-one million protein domains used extensively in bioinformatics, as the pretraining corpus for TAPE. Sequences in Pfam are clustered into evolutionarily-related groups called *families*. We leverage this structure by constructing a test set of fully heldout families (see Section A.5 for details on the selected families), about 1% of the data. For the remaining data we construct training and test sets using a random 95/5% split. Perplexity on the uniform random split test set measures in-distribution generalization, while perplexity on the heldout families test set measures out-of-distribution generalization to proteins that are less evolutionarily related to the training set.

### 4.2 Supervised Datasets

We provide five biologically relevant downstream prediction tasks to serve as benchmarks. We categorize these into structure prediction, evolutionary understanding, and protein engineering tasks. The datasets vary in size between 8 thousand and 50 thousand training examples (see Table S1 for sizes of all training, validation and test sets). Further information on data processing, splits and experimental challenges is in Appendix A.1. For each task we provide:

(**Definition**) A formal definition of the prediction problem, as well as the source of the data.
(**Impact**) The impact of improving performance on this problem.
(**Generalization**) The type of understanding and generalization desired.
(**Metrics**) The metric reported in Table 2 to report results and additional metrics presented in section A.8

**Task 1: Secondary Structure (SS) Prediction (Structure Prediction Task)**

(**Definition**) Secondary structure prediction is a sequence-to-sequence task where each input amino acid $x_i$ is mapped to a label $y_i \in \{\text{Helix}(H), \text{Strand}(E), \text{Other}(C)\}$. See Figure 1a for illustration. The data are from Klausen et al. [34].
(**Impact**) SS is an important feature for understanding the function of a protein, especially if the protein of interest is not evolutionarily related to proteins with known structure [34]. SS prediction tools are very commonly used to create richer input features for higher-level models [35].

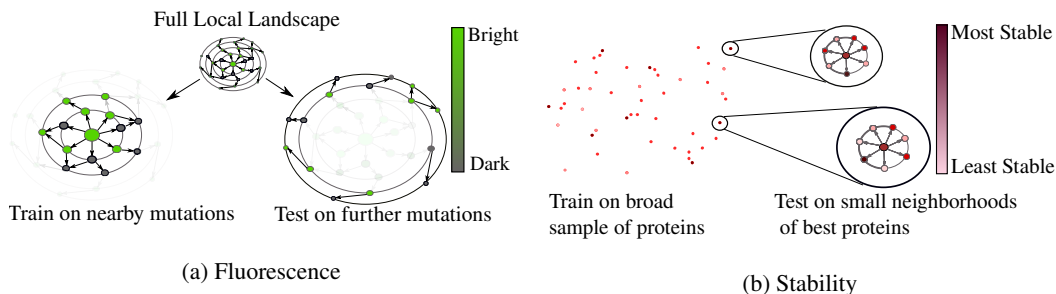

(a) Fluorescence           (b) Stability

Figure 2: Protein Engineering Tasks. In both tasks, a parent protein $p$ is mutated to explore the local landscape. As such, dots represent proteins and directed arrow $x \rightarrow y$ denotes that $y$ has exactly one more mutation than $x$ away from parent $p$. (a) The Fluorescence task consists of training on small neighborhood of the parent green fluorescent protein (GFP) and then testing on a more distant proteins. (b) The Stability task consists of training on a broad sample of proteins, followed by testing on one-mutation neighborhoods of the most promising sampled proteins.

**(Generalization)** SS prediction tests the degree to which models learn local structure. Data splits are filtered at 25% sequence identity to test for broad generalization.

**(Metrics)** We report accuracy on a per-amino acid basis on the CB513 [36] dataset. We further report three-way and eight-way classification accuracy for the test sets CB513, CASP12, and TS115.

### Task 2: Contact Prediction (Structure Prediction Task)

**(Definition)** Contact prediction is a pairwise amino acid task, where each pair $x_i, x_j$ of input amino acids from sequence $x$ is mapped to a label $y_{ij} \in \{0, 1\}$, where the label denotes whether the amino acids are "in contact" ($< 8\text{Å}$ apart) or not. See Figure 1b for illustration. The data are from the ProteinNet dataset [25].

**(Impact)** Accurate contact maps provide powerful global information; e.g., they facilitate robust modeling of full 3D protein structure [37]. Of particular interest are medium- and long-range contacts, which may be as few as twelve sequence positions apart, or as many as hundreds apart.

**(Generalization)** The abundance of medium- and long-range contacts makes contact prediction an ideal task for measuring a model's understanding of global protein context. We select the data splits that was filtered at 30% sequence identity to test for broad generalization.

**(Metrics)** We report precision of the $L/5$ most likely contacts for medium- and long-range contacts on the ProteinNet CASP12 test set, which is a standard metric reported in CASP [24]. We further report Area under PR Curve and Precision at $L$, $L/2$, and $L/5$ for short-range, medium-range and long-range contacts in the supplement.

### Task 3: Remote Homology Detection (Evolutionary Understanding Task)

**(Definition)** This is a sequence classification task where each input protein $x$ is mapped to a label $y \in \{1, \ldots, 1195\}$, representing different possible protein folds. See Figure 1c for illustration. The data are from Hou et al. [38].

**(Impact)** Detection of remote homologs is of great interest in microbiology and medicine; e.g., for detection of emerging antibiotic resistant genes [39] and discovery of new CAS enzymes [40].

**(Generalization)** Remote homology detection measures a model's ability to detect structural similarity across distantly related inputs. We hold out entire evolutionary groups from the training set, forcing models to generalize across large evolutionary gaps.

**(Metrics)** We report overall classification accuracy on the fold-level heldout set from Hou et al. [38]. We further report top-one and top-five accuracy for fold-level, superfamily-level and family-level holdout sets in the supplement.

### Task 4: Fluorescence Landscape Prediction (Protein Engineering Task)

**(Definition)** This is a regression task where each input protein $x$ is mapped to a label $y \in \mathbb{R}$, corresponding to the log-fluorescence intensity of $x$. See Figure 2a for illustration. The data are from Sarkisyan et al. [41].

Table 1: Language modeling metrics: Language Modeling Accuracy (Acc), Perplexity (Perp) and Exponentiated Cross-Entropy (ECE)

| | Random Families | | | Heldout Families | | | Heldout Clans | | |
| --- | --- | --- | --- | --- | --- | --- | --- | --- | --- |
| | Acc | Perp | ECE | Acc | Perp | ECE | Acc | Perp | ECE |
| Transformer | **0.45** | **8.89** | **6.01** | **0.35** | **11.77** | **8.87** | **0.28** | **13.54** | 10.76 |
| LSTM | 0.40 | **8.89** | 6.94 | 0.24 | 13.03 | 12.73 | 0.13 | 15.36 | 16.94 |
| ResNet | 0.41 | 10.16 | 6.86 | 0.31 | 13.19 | 9.77 | **0.28** | 13.72 | **10.62** |
| Bepler et al. [11] | 0.28 | 11.62 | 10.17 | 0.19 | 14.44 | 14.32 | 0.12 | 15.62 | 17.05 |
| Alley et al. [12] | 0.32 | 11.29 | 9.08 | 0.16 | 15.53 | 15.49 | 0.11 | 16.69 | 17.68 |
| Random | 0.04 | 25 | 25 | 0.04 | 25 | 25 | 0.04 | 25 | 25 |

**(Impact)** For a protein of length $L$, the number of possible sequences $m$ mutations away is $O(L^m)$, a prohibitively large space for exhaustive search via experiment, even if $m$ is modest. Moreover, due to epistasis (second- and higher-order interactions between mutations at different positions), greedy optimization approaches are unlikely to succeed. Accurate computational predictions could allow significantly more efficient exploration of the landscape, resulting in better optima. Machine learning methods have already seen some success in related protein engineering tasks [42].

**(Generalization)** The fluorescence prediction task tests the model's ability to distinguish between very similar inputs, as well as its ability to generalize to unseen combinations of mutations. The train set is a Hamming distance-3 neighborhood of the parent green fluorescent protein (GFP), while the test set has variants with four or more mutations. Hamming distance is measured at the amino acid level.

The choice of Hamming distance between amino acids does not always reflect evolution, since not all proteins at the same Hamming distance correspond to equal "evolutionary" distance in the sense of number of nucleotide substitutions. Since we are trying to highlight the protein engineering setting, we believe that this is an important feature of the Fluorescence task. Our goal is to test the models' ability to accurately predict phenotype as a function of an input molecule (e.g. one presented by a protein designer)

**(Metrics)** We report Spearman's $\rho$ (rank correlation coefficient) on the test set. We further report MSE and Spearman's $\rho$ for the full test set, only bright proteins, and only dark proteins in the supplement.

**Task 5: Stability Landscape Prediction (Protein Engineering Task)**

**(Definition)** This is a regression task where each input protein $x$ is mapped to a label $y \in \mathbb{R}$ measuring the most extreme circumstances in which protein $x$ maintains its fold above a concentration threshold (a proxy for intrinsic stability). See Figure 2b for illustration. The data are from Rocklin et al. [43].

**(Impact)** Designing stable proteins is important to ensure, for example, that drugs are delivered before they are degraded. More generally, given a broad sample of protein measurements, finding better refinements of top candidates is useful for maximizing yield from expensive protein engineering experiments.

**(Generalization)** This task tests a model's ability to generalize from a broad sampling of relevant sequences and to localize this information in a neighborhood of a few sequences, inverting the test-case for fluorescence above. The train set consists of proteins from four rounds of experimental design, while the test set contains Hamming distance-1 neighbors of top candidate proteins.

**(Metrics)** We report Spearman's $\rho$ on the test set. In the supplement we also assess classification of a mutation as stabilizing or non-stabilizing. We report Spearman's $\rho$ and accuracy for this task broken down by protein topology in the supplement.

## 5 Models and Experimental Setup

**Losses:** We examine two self-supervised losses that have seen success in NLP. The first is *next-token prediction* [44], which models $p(x_i \mid x_1, \ldots, x_{i-1})$. Since many protein tasks are sequence-to-sequence and require bidirectional context, we apply a variant of next-token prediction which additionally trains the reverse model, $p(x_i \mid x_{i+1}, \ldots, x_L)$, providing full context at each posi-

tion (assuming a Markov sequence). The second is *masked-token prediction* [6], which models $p(x_{\text{masked}} \mid x_{\text{unmasked}})$ by replacing the value of tokens at multiple positions with alternate tokens.

**Protein-specific loss:** In addition to self-supervised algorithms, we explore another protein-specific training procedure proposed by Bepler et al. [11]. They suggest that further *supervised* pretraining of models can provide significant benefits. In particular, they propose supervised pretraining on contact prediction and remote homology detection, and show it increases performance on secondary structure prediction. Similar work in computer vision has shown that supervised pretraining can transfer well to other tasks, making this a promising avenue of exploration [45].

**Architectures and Training:** We implement three architectures: an LSTM [46], a Transformer [47], and a dilated residual network (ResNet) [48]. We use a 12-layer Transformer with a hidden size of 512 units and 8 attention heads, leading to a 38M-parameter model. Hyperparameters for the other models were chosen to approximately match the number of parameters in the Transformer. Our LSTM consists of two three-layer LSTMs with 1024 hidden units corresponding to the forward and backward language models, whose outputs are concatenated in the final layer, similar to ELMo [5]. For the ResNet we use 35 residual blocks, each containing two convolutional layers with 256 filters, kernel size 9, and dilation rate 2. We chose these hyperparameters based on common choices from the literature. Our supervised tasks are of similar size to most of those in the GLUE [49] benchmark, which has been instrumental in demonstrating the success of self-supervision in NLP. Since the models that were applied to GLUE have tens to hundreds of millions of parameters, we chose to make our models roughly the same size. See A.7 for model size ablation experiments. See A.2 for details of how these pretrained models are fed into downstream tasks.

In addition, we benchmark two previously proposed architectures that differ significantly from the three above. The first, proposed by Bepler et al. [11], is a two-layer bidirectional language model, similar to the LSTM discussed above, followed by three 512 hidden unit bidirectional LSTMs. The second, proposed by Alley et al. [12], is a unidirectional mLSTM [50] with 1900 hidden units. Details on implementing and training these architectures can be found in the original papers.

The Transformer and ResNet are trained with masked-token prediction, while the LSTM is trained with next-token prediction. Both Alley et al. and Bepler et al. are trained with next-token prediction. All self-supervised models are trained on four NVIDIA V100 GPUs for one week.

**Baselines:** We evaluate learned features against two baseline featurizations. The first is a one-hot encoding of the input amino acid sequence, which provides a simple baseline. Most current state-of-the-art algorithms for protein classification and regression take advantage of alignment or HMM-based inputs (see Section 2.2). Alignments can be transformed into various features, such as mutation probabilities [38] or the HMM state-transition probabilities [34] for each amino acid position. These are concatenated to the one-hot encoding of the amino acid to form another baseline featurization. For our baselines we use alignment-based inputs that vary per task depending on the inputs used by the current state-of-the-art method. See Appendix A.3 for details on the alignment-based features used for each task. We do not use alignment-based inputs for protein engineering tasks. Proteins in the engineering datasets differ by only a single amino acid, while alignment-based methods search for proteins with high sequence identity, so alignment-based methods return the same set of features for all proteins we wish to distinguish between.

**Experimental Setup:** The goal of our experimental setup is to systematically compare all featurizations. For each task we select a particular supervised architecture, drawing from state-of-the-art where available, and make sure that fine-tuning on all language models is identical. See Appendix A.3 for details on supervised architectures and training.

## 6 Results

Table 1 contains accuracy, perplexity, and exponentiated cross entropy (ECE) on the language modeling task for the five architectures we trained with self-supervision as well as a random model baseline. We report metrics on both the random split and the fully heldout families. Supervised LSTM metrics are reported after language modeling pretraining, but before supervised pretraining. Heldout family accuracy is consistently lower than random-split accuracy, demonstrating a drop in the

Table 2: Results on downstream supervised tasks

| Method | | Structure | | Evolutionary | Engineering | |
|---|---|---|---|---|---|---|
| | | SS | Contact | Homology | Fluorescence | Stability |
| No Pretrain | Transformer | 0.70 | 0.32 | 0.09 | 0.22 | -0.06 |
| | LSTM | 0.71 | 0.19 | 0.12 | 0.21 | 0.28 |
| | ResNet | 0.70 | 0.20 | 0.10 | -0.28 | 0.61 |
| Pretrain | Transformer | 0.73 | 0.36 | 0.21 | **0.68** | **0.73** |
| | LSTM | 0.75 | 0.39 | **0.26** | 0.67 | 0.69 |
| | ResNet | 0.75 | 0.29 | 0.17 | 0.21 | **0.73** |
| | Bepler et al. [11] | 0.73 | 0.40 | 0.17 | 0.33 | 0.64 |
| | Alley et al. [12] | 0.73 | 0.34 | 0.23 | 0.67 | **0.73** |
| Baseline features | One-hot | 0.69 | 0.29 | 0.09 | 0.14 | 0.19 |
| | Alignment | **0.80** | **0.64** | 0.09 | N/A | N/A |

out-of-distribution generalization ability. Note that although some models have lower perplexity than others on both random-split and heldout sets, this lower perplexity does not necessarily correspond to better performance on downstream tasks. This replicates the finding in Rives et al. [29].

Table 2 contains results for all benchmarked architectures and training procedures on all downstream tasks in TAPE. We report accuracy, precision, or Spearman's $\rho$, depending on the task, so higher is always better and each metric has a maximum value of $1.0$. See Section 4 for the metric reported in each task. Detailed results and metrics for each task are in Appendix A.8.

We see from Table 2 that self-supervised pretraining improves overall performance across almost all models and all tasks. Further analysis reveals aspects of these tasks with more for significant improvement. In the fluorescence task, the distribution is bimodal with a mode of bright proteins and a mode of dark proteins (see Figure 3). Since one goal of using machine learning models in protein engineering is to screen potential variants, it is important for these methods to successfully distinguish between beneficial and deleterious mutations. Figure 3 shows that the model does successfully perform some clustering of fluorescent proteins, but that many proteins are still misclassified.

For the stability task, to identify which mutations a model believes are beneficial, we use the parent protein as a decision boundary and label a mutation as beneficial if its predicted stability is higher than the parent's predicted stability. We find that our best pretrained model achieves 70% accuracy in making this prediction while our best non-pretrained model achieves 68% accuracy (see Table S9 for full results). Improving the ability to distinguish beneficial from deleterious mutations would make these models much more useful in real protein engineering experiments.

In the contact prediction task, long-range contacts are of particular interest and can be hundreds of positions apart. Figure 4 shows the predictions of several models on a protein where the longest range contact occurs between the 8th and 136th amino acids. Pretraining helps the model capture more long-range information and improves the overall resolution of the predicted map. However,

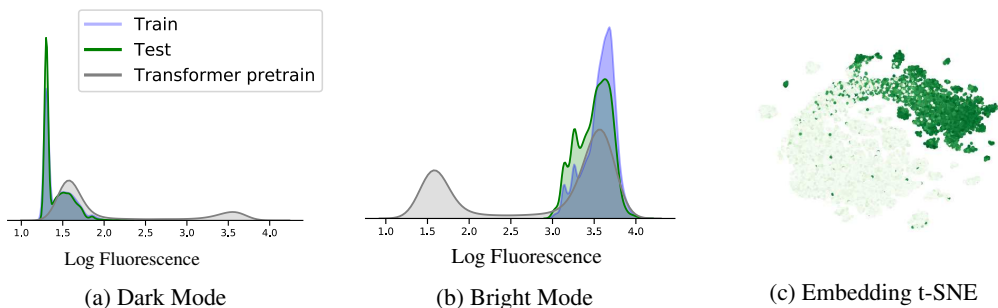

(a) Dark Mode　　　　　(b) Bright Mode　　　　　(c) Embedding t-SNE

Figure 3: Distribution of training, test, and pretrained Transformer predictions on the dark and bright modes, along with t-SNE of pretrained Transformer protein embeddings colored by log-fluorescence.

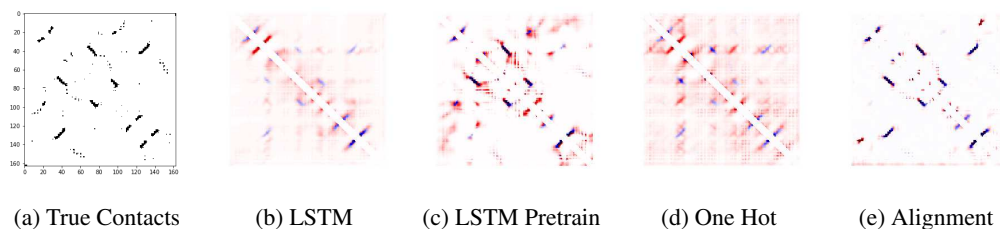

| (a) True Contacts | (b) LSTM | (c) LSTM Pretrain | (d) One Hot | (e) Alignment |

Figure 4: Predicted contacts for chain 1A of a Bacterioferritin comigratory protein (pdbid: 3GKN). Blue indicates true positive contacts while red indicates false positive contacts. Darker colors represent more certainty from the model.

the hand-engineered alignment features result in a much sharper map, accurately resolving more long-range contacts. This increased specificity is highly relevant in structure prediction pipelines [37, 51] and highlights a clear challenge for pretraining.

# 7 Discussion

**Comparison to state of the art.** As shown in Table 2, alignment-based inputs can provide a powerful signal that outperforms current self-supervised models on multiple tasks. Current state-of-the-art prediction methods for secondary structure prediction, contact prediction, and remote homology classification all take in alignment-based inputs. These methods combine alignment-based inputs with other techniques (e.g. multi-task training, kernel regularization) to achieve an additional boost in performance. For comparison, NetSurfP-2.0 [34] achieves 85% accuracy on the CB513 [36] secondary structure dataset, compared to our best model's 75% accuracy, RaptorX [52] achieves 0.69 precision at $L/5$ on CASP12 contact prediction, compared to our best model's 0.49, and DeepSF [38] achieves 41% accuracy on remote homology detection compared to our best model's 26%.

**Need for multiple benchmark tasks.** Our results support our hypothesis that multiple tasks are required to appropriately benchmark performance of a given method. Our Transformer, which performs worst of the three models in secondary structure prediction, performs best on the fluorescence and stability tasks. The reverse is true of our ResNet, which ties the LSTM in secondary structure prediction but performs far worse for the fluorescence task, with a Spearman's $\rho$ of $0.21$ compared to the LSTM's $0.67$. This shows that performance on a single task does not capture the full extent of a trained model's knowledge and biases, creating the need for multi-task benchmarks such as TAPE.

# 8 Future Work

Protein representation learning is an exciting field with incredible room for expansion, innovation, and impact. The exponentially growing gap between labeled and unlabeled protein data means that self-supervised learning will continue to play a large role in machine learning for proteins. Our results show that no single self-supervised model performs best across all protein tasks. We believe this is a clear challenge for further research in self-supervised learning, as there is a huge space of model architecture, training procedures, and unsupervised task choices left to explore. It may be that language modeling as a task is not enough, and that protein-specific tasks are necessary to push performance past state of the art. Further exploring the relationship between alignment-based and learned representations will be necessary to capitalize on the advantages of each. We hope that the datasets and benchmarks in TAPE will provide a systematic model-evaluation framework that allows more machine learning researchers to contribute to this field.

**Acknowledgments** We thank Philippe Laban, David Chan, Jeffrey Spence, Jacob West-Roberts, Alex Crits-Cristoph, Surojit Biswas, Ethan Alley, Mohammed AlQuraishi, Grigory Khimulya, and Kevin Yang for valuable input on this paper. We thank the AWS Educate program for providing us with the resources to train our models. Additionally, we acknowledge funding from Berkeley Deep Drive, Chan-Zuckerberg Biohub, DARPA XAI, NIH, the Packard Fellowship for Science and Engineering, and the Open Philanthropy Project.

## Footnotes

[1]`https://github.com/songlab-cal/tape`

[2]`https://github.com/songlab-cal/tape#citation-guidelines`

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
