[Supplementary Material · Protein_NeurIPS_2019_supplement.pdf]

Table S1: Dataset sizes

| Task | Train | Valid | Test |
|------|-------|-------|------|
| Language Modeling | 32207059 | N/A | 2147130 (Random-split) / 44314 (Heldout families) |
| Secondary Structure | 8678 | 2170 | 513 (CB513) / 115 (TS115) / 21 (CASP12) |
| Contact Prediction | 25299 | 224 | 40 (CASP12) |
| Remote Homology | 12312 | 736 | 718 (Fold) / 1254 (Superfamily) / 1272 (Family) |
| Fluorescence | 21446 | 5362 | 27217 |
| Stability | 53679 | 2447 | 12839 |

# A   Appendix

## A.1   Dataset Details

In Table S1 we show the size of all train, validation, and test sets.

We provide further details about dataset sources, preprocessing decisions, data splitting, and experimental challenges in obtaining labels for each of our supervised tasks below. For ease of reading, each section starts with the following items:

**(Dataset)** The source of the dataset and creation of train/test splits.

**(Labeling)** The current approach to acquiring supervised labels for this task.

### A.1.1   Secondary Structure Details

**(Dataset)** We use a training and validation set from Klausen et al. [31], which is filtered such that no two proteins have greater than 25% sequence identity. We use three test sets, CB513 [33], CASP12 [49], and TS115 [50]. The training set is also filtered at the 25% identity threshold with these test sets. This filtering tests the model's ability to generalize in the interesting case where test proteins are not closely related to train proteins.

**(Labeling)** Determining the secondary structure of a protein experimentally requires high-resolution imaging of the structure, a particularly labor intensive task for structural biologists. Imaging often uses Cryo Electron-Microscopy or X-Ray Crystallography, which can take between weeks and years and can cost over $200,000$ [51].

### A.1.2   Contact Prediction Details

**(Dataset)** We use training, validation, and test sets from ProteinNet [26], which uses a test set based on the CASP12 [49] competition, with training and validation sets filtered at the 30% sequence identity threshold. This tests the ability of the model to generalize to proteins that are not closely related to any train proteins.

**(Labeling)** Determining the contacts of a protein requires knowing its full 3D structure. As with secondary structure, determining the 3D structure requires imaging a protein.

### A.1.3   Remote Homology Details

**(Dataset)** We use a training, validation, and test set from [35], derived from the SCOP 1.75 database [52] of hierarchically classified protein domains. All proteins of a given fold are further categorized into related *superfamilies*. Entire superfamilies are held out from the training set, allowing us to evaluate how the model generalizes across evolutionary distance when structure is preserved.

**(Labeling)** Each fold is annotated from the structure of the sequence, which SCOP pulls from the Protein DataBank [52, 53]. Finding new superfamilies with the same fold is a challenging task, requiring sequencing in extreme environments as is often done in metagenomics [54].

### A.1.4   Fluorescence Details

**(Dataset)** We use data generated by an experimental technique called Deep Mutational Scanning (DMS) [38]. This technique allows for extensive characterizations of small neighborhoods of a parent protein through mutagenesis. We create training, validation, and test splits ourselves,

504     partitioning the data so that train and validation are in a Hamming distance 3 neighborhood of the
505     original protein, while test data is a sample from the Hamming distance 4-15 neighborhood.
506 **(Labeling)** DMS is efficient for local characterization near a single protein, but its samples become
507     vanishingly small once neighborhoods start to expand outside of Hamming distance 2.

### A.1.5 Stability Details

509 **(Dataset)** We use data generated by a novel combination of parallel DNA synthesis and protein
510     stability measurements [40]. We create training, validation, and test splits ourselves, partitioning
511     the data so that training and validation sets come from four rounds of experimental data measuring
512     stability for many candidate proteins, while our test set consists of seventeen 1-Hamming distance
513     neighborhoods around promising proteins observed in the four rounds of experimentation.
514 **(Labeling)** This approach for observing stability is powerful because of its throughput, allowing the
515     authors to find the most stable proteins ever observed for certain classes [40]. The authors observe
516     that the computational methods used to guide their selection at each stage could be improved,
517     meaning that in this case better models could actually lead to better labeled data in a virtuous cycle.

## A.2 Supervised Architectures

519 For each task, we fixed one supervised architecture and tried one-hot, alignment-based, and neural net
520 based features. We did not perform hyperparameter tuning or significant architecture optimization, as
521 the main goal was to compare feature extraction techniques.

522 For each task we define the supervised architecture below. If this is a state of the art architecture from
523 other work, we highlight any novel training procedure or inputs they take.

### A.2.1 Secondary Structure Architecture

525 We used the NetSurfP2.0 model from Klausen et al [31]. The model consists of two convolutional
526 layers followed by two bidirection LSTM layers and a linear output layer. The convolutional layers
527 have filter size 32 and kernel size 129 and 257, respectively. The bidirectional LSTM layers have
528 1024 hidden units each.

529 In the original model, the authors take in the outputs of an HMM-HMM alignment method called
530 HHblits [14] in addition to a one-hot encoding of the sequence, giving $50$-dimensional inputs at each
531 position. They train the model on multiple tasks including secondary structure prediction (3 and 8
532 class), bond-angle prediction, and solvent accessibility prediction. For clarity, we only compared
533 to the model trained without the multitask training, which in our experiments contributed an extra
534 one to two percent in test accuracy. In addition to multitask training, they balance the losses between
535 different tasks to achieve maximum accuracy on secondary structure prediction. All features and
536 code to do the full multitask training is available in our repository.

### A.2.2 Contact Prediction Architecture

538 We used a supervised network inspired by the RaptorX-Contact model from Ma et al [48]. Since a
539 contact map is a 2D pairwise prediction, we form a 2D input from our 1D features by concatenating
540 the features at position $i$ and $j$ for all $i, j$. This 2D input is then passed through a convolutional
541 residual network with. The 2D network contains 30 residual blocks with two convolutional layers
542 each. Each convolution in the residual block has filter size 64 and a kernel size of 3.

543 The original RaptorX method inputs uses alignment-based methods to find similar proteins, then
544 passes these through CCMpred [55] - a Markov Random Field based contact prediction method. This
545 outputs a 2D featurization including mutual information and pairwise potential. This, along with
546 1D HMM alignment features and the one-hot encoding of each amino acid are fed their network.
547 Unfortunately the code and features are not publically available, so we used the 1D alignment-based
548 features available in ProteinNet [26] instead. While this improved performance significantly, the
549 numbers reported by RaptorX are higher than those we obtained with our implementation.

### A.2.3 Remote Homology Architecture

551 Remote homology requires a single prediction for each protein. To obtain a sequence-length invariant
552 protein embedding we compute an attention-weighted mean of the amino acid embeddings. More

precisely, we predict an attention value for each position in the sequence using a trainable dense layer, then use those attention values to compute an attention-weighted mean protein embedding. This protein embedding is then passed through a 512 hidden unit dense layer, a relu nonlinearity, and a final linear output layer to predict logits for all 1195 classes. We note that Hou et al. [35] propose a deep architecture for this task and report state of the art performance. When we compared the performance of this supervised architecture to that of the attention-weighted mean above, the attention-based embedding performed better for all featurizations. As such, we choose to report results using the simpler attention-based downstream architecture.

The current state of the art method in this problem, DeepSF [35], takes in a one-hot encoding of the amino acids, predicted secondary structure labels, predicted solvent accessibility labels, and a 1D alignment-based features. In an ablation study, the authors show that the secondary structure labels are most useful for performance of their model. We report only one-hot and alignment-based results in the main paper to maintain consistency with alignment-based featurizations for other tasks. All input features used by DeepSF are available in our repository.

### A.2.4   Protein Engineering Architectures

Protein engineering also requires a single prediction for each protein. Therefore we use the same architecture as we do for remote homology, computing an attention-weighted mean protein embedding, a dense layer with 512 hidden units, a relu nonlinearity and a final linear output layer to predict the quantity of interest (either stability or fluorescence).

Since we create these training, validation, and test splits ourselves, no clear previous state of the art exists. Related work on protein engineering has used a similar architecture by computing a single protein embedding followed by some form of projection (linear or with a small feed forward network) [12, 28]. These methods also do not take in alignment-based features and only use one-hot amino acids as inputs.

### A.3   Training Details

Self-supervised models were all trained on four NVIDIA V100 GPUs on AWS for 1 week. Training used a learning rate of $10^{-3}$ with a linear warm up schedule, the Adam optimizer, and a 10% dropout rate. Since proteins vary in length significantly, we use variable batch sizes depending on the length of the protein. These sizes also differ based on model architecture, as some models (e.g. the Transformer) have significantly higher memory requirements. Specific batch sizes for each model at each protein length are available in our repository.

Supervised models were trained on two GPUs until convergence (no increase in validation accuracy for 10 epochs) with the exception of the memory-intensive Contact Prediction task, which was trained on four GPUs until convergence. Training used a learning rate of $10^{-4}$ with a linear warm up schedule, the Adam optimizer, and a 10% dropout rate. We backpropagated fully through all models during supervised fine-tuning.

In addition, due to high memory requirements of some downstream tasks (especially contact prediction) we use memory saving gradients [56] to fit more examples per batch on the GPU.

### A.4   Pfam Heldout Families

The following Pfam clans were held out during self-supervised training: CL0635, CL0624, CL0355, CL0100, CL0417, CL0630. The following Pfam families were held out during self-supervised training: PF18346, PF14604, PF18697, PF03577, PF01112, PF03417. First, a "clan" is a cluster of families grouped by the maintainers of Pfam based on shared function or evolutionary origin (see [30] for details). We chose holdout clans and families in pairs, where a clan of novel function is held out together with a family that is similar in sequence but different evolutionarily or functionally. This serves to simultaneously test generalization across large distances (entirely held out families) and between similar looking unseen groups (e.g. the paired holdout clan and holdout family).

Table S2: Detailed secondary structure results

| | | Three-Way Accuracy (Q3) | | | Eight-Way Accuracy (Q8) | | |
| | | CB513 | CASP12 | TS115 | CB513 | CASP12 | TS115 |
|---|---|---|---|---|---|---|---|
| No Pretrain | Transformer | 0.70 | 0.68 | 0.73 | 0.51 | 0.52 | 0.58 |
| | LSTM | 0.71 | 0.69 | 0.74 | 0.47 | 0.48 | 0.52 |
| | ResNet | 0.70 | 0.68 | 0.73 | 0.55 | 0.56 | 0.61 |
| Pretrain | Transformer | 0.73 | 0.71 | 0.77 | 0.59 | 0.59 | 0.64 |
| | LSTM | 0.75 | 0.70 | 0.78 | 0.59 | 0.57 | 0.66 |
| | ResNet | 0.75 | 0.72 | 0.78 | 0.58 | 0.58 | 0.64 |
| Supervised [11] | LSTM | 0.73 | 0.70 | 0.76 | 0.58 | 0.57 | 0.65 |
| UniRep [12] | mLSTM | 0.73 | 0.72 | 0.77 | 0.57 | 0.59 | 0.63 |
| Baseline | One-hot | 0.69 | 0.68 | 0.72 | 0.52 | 0.53 | 0.58 |
| | Alignment | **0.8** | **0.76** | **0.81** | **0.63** | **0.61** | **0.68** |

## A.5    Bepler Supervised Training

We perform supervised pretraining using the same architecture described in Bepler et al. [11]. We train on the same tasks, a paired remote homology task and contact map prediction task. However, in order to accurately report results on downstream secondary structure, contact map, and remote homology datasets, which were filtered by sequence identity, we perform this same sequence identity filtering on the supervised pretraining set. This reduced the supervised pretraining dataset size by 75% which likely reduced the effectiveness of the supervised pretraining. Both filtered and unfiltered supervised pretraining datasets are made available in our repository.

## A.6    Detailed Results on Supervised Tasks

Here we provide detailed results on each task, examining multiple metrics and test-conditions to further determine what the models are learning.

### A.6.1    Secondary Structure Results

We perform both three-class and eight-class secondary structure classification following the DSSP labeling system [57]. Three way classification tags each position as either Helix, Strand or Other. Eight-way classification breaks these three labels into more specialized classes, for example Helix is broken into 3-turn, 4-turn or 5-turn helix. Table S2 shows results on these tasks. We note that test-set performance is comparable for all three test sets, in particular alignment does better at both eight-way and three-way classification by a large margin.

We follow the standard notation, where Q3 refers to three-way classification accuracy and Q8 refers to eight-way classification accuracy.

### A.6.2    Contact Prediction Results

We report all metrics commonly used to capture contact prediction results [48] in tables S4 and S5. The metrics "P@K" are precision for the top $K$ contacts, where all contacts are sorted from highest confidence to lowest confidence. Note that $L$ is the length of the protein, so "P@L/2", for example, denotes the precision for the $L/2$ most likely predicted contacts in a protein of length $L$. In Table S4 we report all metrics for medium range contacts, which are contacts for positions between five and twelve amino acids apart. In Table S5 we report all metrics for long range contacts, which are contacts for positions greater than 12 amino acids apart.

All results decay as we transition from short range to long range contacts, which we note is *not* the case for many state of the art methods from recent CASP competitions [47, 48]. That said, for long-range contacts, the pretrained LSTM does report higher precision for top predictions than the simple alignment-based features we use here, highlighting the potential for further gains in the future.

Table S3: Detailed short-range contact prediction results. Short range contacts are contacts between positions separated by 6 to 11 positions, inclusive.

|  |  | Pr | Recall | F1 | AUPRC | P@L | P@L/2 | P@L/5 |
|---|---|---|---|---|---|---|---|---|
| No Pretrain | Transformer | 0.05 | 0.93 | 0.09 | 0.11 | 0.1 | 0.12 | 0.13 |
|  | LSTM | 0.05 | 0.93 | 0.09 | 0.29 | 0.23 | 0.3 | 0.38 |
|  | ResNet | 0.05 | 0.93 | 0.09 | 0.27 | 0.21 | 0.28 | 0.36 |
| Pretrain | Transformer | 0.05 | 0.93 | 0.09 | 0.3 | 0.23 | 0.3 | 0.38 |
|  | LSTM | 0.05 | 0.93 | 0.09 | 0.39 | 0.27 | 0.36 | 0.49 |
|  | ResNet | 0.05 | 0.74 | 0.08 | 0.22 | 0.18 | 0.23 | 0.3 |
| Supervised [11] | LSTM | 0.05 | 0.93 | 0.09 | 0.33 | 0.25 | 0.33 | 0.43 |
| UniRep [12] | mLSTM | 0.05 | 0.92 | 0.09 | 0.36 | 0.25 | 0.33 | 0.43 |
| Baseline | One-hot | 0.05 | 0.93 | 0.09 | 0.28 | 0.22 | 0.29 | 0.36 |
|  | Alignment | 0.05 | **1.0** | **0.10** | **0.51** | **0.35** | **0.5** | **0.66** |

Table S4: Detailed medium-range contact prediction results. Medium range contacts are contacts between positions separated by 12 to 23 positions, inclusive.

|  |  | Pr | Recall | F1 | AUPRC | P@L | P@L/2 | P@L/5 |
|---|---|---|---|---|---|---|---|---|
| No Pretrain | Transformer | 0.03 | 0.88 | 0.06 | 0.07 | 0.07 | 0.07 | 0.09 |
|  | LSTM | 0.03 | 0.88 | 0.06 | 0.2 | 0.16 | 0.2 | 0.27 |
|  | ResNet | 0.03 | 0.88 | 0.06 | 0.18 | 0.14 | 0.18 | 0.23 |
| Pretrain | Transformer | 0.03 | 0.88 | 0.06 | 0.19 | 0.16 | 0.2 | 0.25 |
|  | LSTM | 0.03 | 0.88 | 0.06 | 0.31 | 0.21 | 0.29 | 0.39 |
|  | ResNet | 0.03 | 0.69 | 0.06 | 0.15 | 0.11 | 0.15 | 0.2 |
| Supervised [11] | LSTM | 0.03 | 0.88 | 0.06 | 0.26 | 0.19 | 0.25 | 0.33 |
| UniRep [12] | mLSTM | 0.03 | 0.87 | 0.06 | 0.29 | 0.2 | 0.26 | 0.34 |
| Baseline | One-hot | 0.03 | 0.88 | 0.06 | 0.18 | 0.15 | 0.18 | 0.23 |
|  | Alignment | 0.03 | **0.98** | 0.06 | **0.45** | **0.32** | **0.45** | **0.59** |

Table S5: Detailed long-range contact prediction results. Long range contacts are contacts between positions separated by 24 or more positions, inclusive.

|  |  | Pr | Recall | F1 | AUPRC | P@L | P@L/2 | P@L/5 |
|---|---|---|---|---|---|---|---|---|
| No Pretrain | Transformer | 0.02 | 0.87 | 0.03 | 0.05 | 0.07 | 0.09 | 0.11 |
|  | LSTM | 0.02 | 0.87 | 0.03 | 0.11 | 0.16 | 0.2 | 0.25 |
|  | ResNet | 0.02 | 0.86 | 0.03 | 0.1 | 0.14 | 0.19 | 0.24 |
| Pretrain | Transformer | 0.02 | 0.87 | 0.03 | 0.11 | 0.17 | 0.21 | 0.26 |
|  | LSTM | 0.02 | 0.87 | 0.03 | 0.2 | **0.26** | **0.32** | **0.39** |
|  | ResNet | 0.02 | 0.61 | 0.03 | 0.09 | 0.14 | 0.17 | 0.21 |
| Supervised [11] | LSTM | 0.02 | 0.87 | 0.03 | 0.14 | 0.22 | 0.27 | 0.33 |
| UniRep [12] | mLSTM | 0.02 | 0.85 | 0.03 | 0.18 | 0.24 | 0.29 | 0.35 |
| Baseline | One-hot | 0.02 | 0.87 | 0.03 | 0.08 | 0.12 | 0.15 | 0.2 |
|  | Alignment | 0.02 | 0.86 | 0.03 | **0.15** | 0.23 | 0.29 | 0.35 |

Table S6: Detailed remote homology prediction results

| | | Fold | | Superfamily | | Family | |
|---|---|---|---|---|---|---|---|
| | | Top-1 | Top-5 | Top-1 | Top-5 | Top-1 | Top-5 |
| No Pretrain | Transformer | 0.09 | 0.21 | 0.07 | 0.2 | 0.31 | 0.58 |
| | LSTM | 0.12 | 0.28 | 0.13 | 0.29 | 0.68 | 0.85 |
| | ResNet | 0.1 | 0.24 | 0.07 | 0.19 | 0.39 | 0.6 |
| Pretrain | Transformer | 0.21 | 0.37 | 0.34 | 0.51 | 0.88 | 0.94 |
| | LSTM | **0.26** | **0.43** | **0.43** | **0.59** | **0.92** | **0.97** |
| | ResNet | 0.17 | 0.29 | 0.31 | 0.44 | 0.77 | 0.87 |
| Supervised [11] | LSTM | 0.17 | 0.30 | 0.20 | 0.36 | 0.79 | 0.91 |
| UniRep [12] | mLSTM | 0.23 | 0.39 | 0.38 | 0.54 | 0.87 | 0.94 |
| Baseline | One-hot | 0.09 | 0.21 | 0.08 | 0.21 | 0.39 | 0.66 |
| | Alignment | 0.09 | 0.21 | 0.09 | 0.24 | 0.53 | 0.77 |

Table S7: Detailed fluorescence prediction results. $\rho$ denotes Spearman $\rho$.

| | | Full Test Set | | Bright Mode Only | | Dark Mode Only | |
|---|---|---|---|---|---|---|---|
| | | MSE | $\rho$ | MSE | $\rho$ | MSE | $\rho$ |
| No Pretrain | Transformer | 2.59 | 0.22 | 0.08 | 0.08 | 3.79 | 0 |
| | LSTM | 2.35 | 0.21 | 0.11 | 0.05 | 3.43 | -0.01 |
| | ResNet | 2.79 | -0.28 | **0.07** | -0.07 | 4.1 | -0.01 |
| Pretrain | Transformer | 0.22 | **0.68** | 0.09 | 0.60 | 0.29 | **0.05** |
| | LSTM | **0.19** | 0.67 | 0.12 | 0.62 | **0.22** | 0.04 |
| | ResNet | 3.04 | 0.21 | 0.12 | 0.05 | 4.45 | 0.02 |
| Supervised [11] | LSTM | 2.17 | 0.33 | 0.08 | 0.06 | 3.17 | 0.02 |
| UniRep [12] | mLSTM | 0.20 | 0.67 | 0.13 | **0.63** | 0.24 | 0.04 |
| Baseline | One-hot | 2.69 | 0.14 | 0.08 | 0.03 | 3.95 | 0.0 |

### A.6.3 Remote Homology Results

In Table S6, we report results on three remote homology test datasets constructed in Hou et al [35]. Recall that "Fold" has the most distantly related proteins from train, while "Superfamily" and "Family" are increasingly related (see Section A.1.3 for more details). This is reflected in the accuracies in Table S6, which increase drastically as the test sets get easier.

### A.6.4 Fluorescence Results

Fluorescence distribution in the train, validation, and test sets is bimodal, with one mode corresponding to bright proteins and one mode corresponding to dark proteins. The dark mode is significantly more diverse in the test set than the train and validation sets, which makes sense as most random mutations will destroy the refined structure necessary for fluorescence. With this in mind, we report Spearman's $\rho$ and mean-squared-error (MSE) on the whole test-set, on only dark mode, and on only the bright mode in Table S7. The drop in MSE for both modes shows that pretraining helps our best models distinguish between dark and bright proteins. However low Spearman's $\rho$ for the dark mode suggests that models are not able to rank proteins within this mode.

Table S8: Overall stability prediction results

|  |  | Spearman's $\rho$ | Accuracy |
|---|---|---|---|
| No Pretrain | Transformer | -0.06 | 0.5 |
|  | LSTM | 0.28 | 0.6 |
|  | ResNet | 0.61 | 0.68 |
| Pretrain | Transformer | **0.73** | **0.70** |
|  | LSTM | 0.69 | 0.69 |
|  | ResNet | **0.73** | 0.66 |
| Supervised [11] | LSTM | 0.64 | 0.67 |
| UniRep [12] | mLSTM | **0.73** | 0.69 |
| Baseline | One-hot | 0.19 | 0.58 |

## A.6.5 Stability Results

Table S9: Stability prediction results broken down by protein topology

|  |  | $\alpha\alpha\alpha$ | | $\alpha\beta\beta\alpha$ | | $\beta\alpha\beta\beta$ | | $\beta\beta\alpha\beta\beta$ | |
|---|---|---|---|---|---|---|---|---|---|
|  |  | $\rho$ | Acc | $\rho$ | Acc | $\rho$ | Acc | $\rho$ | Acc |
| No Pretrain | Transformer | -0.39 | 0.49 | -0.41 | 0.47 | 0.52 | 0.5 | 0.25 | 0.52 |
|  | LSTM | -0.07 | 0.57 | 0.39 | 0.7 | -0.43 | 0.56 | -0.34 | 0.56 |
|  | ResNet | 0.64 | 0.69 | 0.16 | 0.69 | 0.63 | 0.67 | 0.65 | 0.67 |
| Pretrain | Transformer | 0.66 | 0.68 | **0.48** | 0.73 | 0.65 | **0.71** | 0.65 | 0.67 |
|  | LSTM | 0.71 | **0.7** | 0.17 | 0.73 | **0.68** | 0.67 | **0.67** | **0.7** |
|  | ResNet | 0.68 | 0.68 | 0.15 | 0.63 | 0.61 | 0.68 | 0.6 | 0.68 |
| Supervised [11] | LSTM | 0.33 | 0.66 | 0.24 | **0.79** | 0.54 | 0.7 | 0.58 | 0.53 |
| UniRep [12] | mLSTM | **0.72** | 0.66 | 0.11 | 0.76 | **0.68** | 0.66 | 0.65 | 0.67 |
| Baseline | One-hot | 0.58 | 0.59 | 0.04 | 0.58 | -0.05 | 0.58 | 0.54 | 0.58 |

The goal of the Rocklin et al. [40] experiment was to find highly stable proteins. In the last stage of this experiment they examine variants of the the most promising candidate proteins. Therefore we wish to measure both whether our model was able to learn the landscape around these candidate proteins, as well as whether it successfully identified those variants with greater stability than the original parent proteins. In Table S8 we report Spearman's $\rho$ to measure the degree to which the landscape was learned. In addition, we report classification accuracy of whether a mutation is beneficial or harmful using the predicted stability of the parent protein as a decision boundary.

In Table S9 report all metrics separately for each of the four protein topologies tested in Rocklin et al [40], where $\alpha$ denotes a helix and $\beta$ denotes a strand (or $\beta$-sheet). We do this because success rates varied significantly by topology in their experiments, so some topologies (such as $\alpha\alpha\alpha$ were much easier to optimize than others (such as $\alpha\beta\beta\alpha$). We find that our prediction success also varies significantly by topology.