[Reviews · NeurIPS 2019]

Reviewer 1



Comments: - The paper is incredibly well written and very concise (one of the clearest papers I have reviewed). - In Table 2, do the results corresponding to [11] and [12] involve pre-training? If not, it would make them the best architectures without pre-training. I wonder if pre-training would further improve their performance. - It would be nice to have a slightly more detailed explanation of the alignment-based features used in this work. - Does pre-training help if you use alignment-based features for the secondary structure and contact prediction problems? Right now, it is unclear which rows involve models with pre-training and which rows do not. - A further investigation of why pre-training reduces the performance for the contact prediction problem would be interesting.

Reviewer 2



The manuscript presents a set of diverse protein prediction tasks, with the purpose of establishing a benchmark for testing representation/transfer learning on protein sequence data. In addition, it establishes a strong baseline for the field by implementing a range of different standard sequence models, and demonstrating their performance on a benchmark set. I expect both the benchmark set, and the results reported in this paper to have a substantial impact on the community. Below are some comments and suggestions for changes Page 3. Since the goal is to "ensure that no test proteins are closely related to train proteins", it would be informative if the authors could state the expected (or maximum) sequence identity between PFAM families. Wouldn't it have made sense to do the split at the clan level, to reduce the chance of information leakage between families within the same superfamily? About task 2: much of recent progress in protein structure prediction comes from prediction of distance distributions rather than a simple binary classification of contact presence. You could perhaps consider modifying task 2 to this more challenging problem, as it is closer to a real-world application. I realize that this is not feasible within the time-frame of a NeurIPS rebuttal, so this is merely a suggestion for future development of the benchmark. About task 4: Is the Hamming distance by which you measure mutation-degree at the nucleotide or the amino acid level? As far as I can see, both would have subtle problems. If it is done at the nucleotide level, then the exact same amino acid sequence might end up in both train and test set (due to synonymous mutations). If it is measured at the amino acid level, then not all mutations at the same Hamming-distance would be equally distant. The authors should clarify this. Page 6, line 208 The authors write "(Assuming a Markov sequence)", however, it is not clear to me that such an assumption is actually made, since the model is conditioned on all preceding amino acids. Perhaps I missed it, but it was not entirely clear to me which representation was used as the basis for the different downstream tasks. For instance, for the LSTM, do the authors simply use the 2x1024 hidden unit state at each position as feature representation? Doesn't this mean that the dimension of the representation is different for the different methods. Have the authors investigated whether the difference in representation size alone has an effect on downstream performance (for instance, one would imagine that the downstream model would need a larger amount of parameters if the input dimensions were larger). Furthermore, for the tasks where a single prediction is made for an entire protein, how are the individual representations combined? Is this done by averaging over the individual representations, or by some more complex scheme (such as the averaging + last-state used by UniRep for some of its applications). Quality: The work is systematic and carefully executed. Apart from the questions above, I have full confidence that the submission is technically sound. Clarity: The manuscript is well structured and well written, and emphasis has been placed on making it accessible to a broader ML community. I have a few minor comments that could improve the clarify even further 1. Figure 1 (a). The blue and yellow strand cartoon next to the input and output are confusing. The yellow should be removed. Furthermore, the E->C transition is not clearly visible in the chosen cartoon representation. 2. Figure 1 (b). There seem to be some contacts missing between the first and last strand in the sequence. There is a single points, but I assume there should be an entire strand. This is certainly a detail, but for the sake of clarify it would be good to get this right. 3. Page 4. Line 143. Perhaps add (H), (E), and (C) labels after the full names in {Helix, Strand, Other}, just to make the coupling to Fig 1a clearer. 4. There seems to be a word missing in line 262: "tasks with more for significant improvement". Originality: The submitted work is original, and clearly cites earlier work. Significance: There is currently a lot of activity in semi-supervised and unsupervised learning of biological sequences. It has, however, been difficult to assess the extent of the progress that these approaches constitute over earlier methods. This current work represents a real step forward in this direction, by establishing a solid benchmark, and by clearly demonstrating the areas in which the learned representations still underperform classic Bioinformatics approaches. I expect it to have a substantial impact on the field. Updated after rebuttal: the authors have addressed my concerns. I have therefore updated my score to 8.

Reviewer 3



This manuscript assembles and describes a set of benchmark data sets for use by the machine learning community in evaluating semi-supervised learning in relation to various properties of proteins. The authors also demonstrate the application of several self-supervised pretraining methods to these tasks. The manuscript is extremely clearly written, which is critical in work for which one of the primary goals is communicating to non-specialists in computational biology. The benchmarks are well constructed, though the actual work involved in curating them was not particularly substantial, since most of the benchmarks have been previously published and are merely collated in this work into one collection. Still, the utility is clear, since practitioners can now go to one place and test their methods using a single interface on a variety of tasks. In terms of modeling, the approaches they use represent architectures that frequently yield state-of-the-art performance in NLP or computer vision, such as transformers or ResNets. One concern is that the models employed here are huge (on the order of 38M parameters). The manuscript should provide some justification for using such large models. The concern here is two-fold: (1) these complex models may not be as powerful as simpler models, especially on some of the smaller benchmarks, and (2) beginning with massive models that require significant compute resources may discourage potential users of the benchmarks. A significant problem with all of the benchmarks is the simplicity of the evaluation, where only global accuracy metrics are considered. Methods like precision-recall and ROC analysis are frequently employed in this domain. It might also be helpful to explore metrics such as accuracy at predicting long range contacts or error partitioned by the true label. Note that this would mean evaluating accuracy for low stability and high stability proteins separately. line 150: The CB513 data set is quite standard in secondary structure prediction, but it is also quite old, dating from 1999. A newer, larger benchmark should be employed. Minor comments: line 1: "Protein modeling" is vague. Clarify what task is being addressed here by including information from lines 37-38. line 75: The term "homology" is reserved only for proteins that share a common evolutionary ancestor. If the proteins share similar function but no common ancestor, they are not homologous. line 125: The text says that the test set is constructed by holding out families, but in the next sentence "For the remaining data we construct training and test sets using a random 95%/5% split." This makes no sense. line 134: "8 thousand" -> "8000"

[Author Response · NeurIPS 2019]

We thank the reviewers for their positive feedback and helpful suggestions for improvement.

**Response to Reviewer 1** Both Bepler et al. [11] and Alley et al. [12] were pretrained on the PFAM corpus. We have updated our table to make this clearer. We have expanded the discussion of alignment-based features in the Background section to make it clearer what they are. Since alignment-based features are fed directly into the downstream task models, this is separate from pretraining on the language modeling tasks.

The reviewer points out the unexpected result that pretraining decreased contact prediction performance with the ResNet. This was caused by an issue with our training/evaluation setup for contact map prediction, which we discovered after the submission deadline. When we re-ran training/evaluation for all models, we found that pretraining improved accuracy across the board, as shown in Table 1.

| Method | Contact |
|---|---|
| Transformer (No PT) | 0.32 |
| LSTM (No PT) | 0.19 |
| ResNet (No PT) | 0.20 |
| Transformer (PT) | 0.36 |
| LSTM (PT) | 0.39 |
| ResNet (PT) | 0.29 |
| Bepler (PT) | 0.40 |
| UniRep (PT) | 0.34 |

Table 1: Updated Contact Results

**Response to Reviewer 2** When splitting PFAM, we split at both the family and clan level. We will present these results separately in Table 1 of the paper. We ran BLAST to obtain sequence identity between these sets. We find that that on average the maximum sequence identity between the training set and proteins in the random-split set is 73%, in the family-holdout is 37%, and in the clan-holdout is 22%. Hamming distance is measured at the amino acid level, which we chose because we are trying to highlight the protein engineering setting where the goal is to accurately predict the function of an input molecule specified by a protein designer. We have added this reasoning to the task description.

The reviewer is correct that the LSTM language model is not making a Markovian assumption, and we have removed this line. The reviewer is correct in their characterization of the LSTM, which is the concatenation of the two vectors of dimension 1024. For the single-prediction tasks, we use a learned attention-weighted sum over the amino-acid representations. These points are in the Appendix, and we have added pointers to these details in the main text.

We agree that difference in representation size could make a difference on downstream tasks. Due to cost of pretraining large models, we cannot perform a grid search over number of parameters and representation size for pretrained models. As such, we instead chose to base models on common hyperparameter choices in the literature. In addition, controlling for number of parameters and representation size proved difficult without resorting to extremely odd choices or significant alterations to the network design. For example, lowering the LSTM hidden dimension to 512 requires 10 layers to match the parameters in the Transformer. To address these concerns, we are pretraining three smaller models with a fixed representation size to compare architectures with representation size held constant (see Table 2).

**Response to Reviewer 3** We agree with the reviewer that the language models we trained are quite large, which may present a problem for fine-tuning on small datasets. Our tasks are of similar size to most of those in the GLUE benchmark, which has been instrumental in demonstrating the success of self-supervision in NLP. Since the models that were applied to GLUE have tens to hundreds of millions of parameters, we chose to make our models roughly the same size ($\sim$40M parameters). We also believe that showing results of larger models is important to counter the claim that simply scaling these models would allow them to outperform gold-standard features from bioinformatics. We have added this discussion in the manuscript. To further address questions about size, we are pretraining smaller language models with $\sim$3M parameters for a week on Pfam. Some preliminary results on the Transformer and LSTM after two days of pretraining are reported in Table 2.

| Method | SS | Stab | Fluor | RH |
|---|---|---|---|---|
| LSTM | 0.73 | 0.67 | 0.66 | 0.18 |
| Transformer | 0.70 | 0.68 | 0.68 | 0.13 |
| ResNet | 0.73 | 0.68 | 0.43 | 0.11 |

Table 2: Small Language Model Results

We agree with the reviewer that usability of the benchmark is of paramount concern. The larger models require around 11 GB of memory on a GPU to train, which is on the higher end. To alleviate this, we will upload weights for the smaller models once they are trained.

In Table 2 of the paper we chose to report simple global metrics. However, we report many detailed metrics in the appendix, including some of the metrics the reviewer suggested: precision-recall, AUPRC, and accuracy for long-range contacts. For the other tasks we provided label/task-specific breakdowns in the appendix, and have added more pointers to the appendix to improve the visibility of these results. We have also revised the discussion in the appendix for ease of reading. We also plan to create an online leaderboard, where these other metrics will be more prominent. We agree with the reviewer that CB513 is built from an old dataset and will look into improving the test set for future iterations of TAPE.

**Clarity** We thank all reviewers for positive comments on clarity. All small edits and suggested improvements have been incorporated. In particular we are grateful for Reviewer 2's catch of the missing beta strand contacts (the contact map was cropped incorrectly, and has been updated). Additionally, as per Reviewer 3's suggestion, we have changed 'Protein Modeling' to 'Machine learning applied to protein sequences' in the abstract and throughout the manuscript.

[Meta-Review · NeurIPS 2019]

The contributions of this paper are multi-dimensional and highly significant: (i) developing a set of benchmarks for a diverse prediction tasks, (ii) demonstrating the utility of incorporating the vast amount of unlabeled protein data to pre-train models via semi-supervised learning, and (iii) the unlabeled data and pre-trained models made publicly available. This work will make a significant impact on the field by establishing solid benchmarks and facilitate the introduction of challenging protein prediction tasks to the machine learning community. The paper is extremely clearly written, well-structured and very concise. All reviewers are satisfied by the author response.